# TAQing2.0 for genome reorganization of asexual industrial yeasts by direct protein transfection

Taishi Yasukawa[1,4], Arisa H. Oda [2,4], Takahiro Nakamura[2,4], Naohisa Masuo[1], Miki Tamura[2], Yuriko Yamasaki[1], Makoto Imura[1], Takatomi Yamada[2] & Kunihiro Ohta [2,3✉]

Genomic rearrangements often generate phenotypic diversification. We previously reported the TAQing system where genomic rearrangements are induced via conditional activation of a restriction endonuclease in yeast and plant cells to produce mutants with marked phenotypic changes. Here we developed the TAQing2.0 system based on the direct delivery of endonucleases into the cell nucleus by cell-penetrating peptides. Using the optimized procedure, we introduce a heat-reactivatable endonuclease TaqI into an asexual industrial yeast (torula yeast), followed by a transient heat activation of TaqI. TAQing2.0 leads to generation of mutants with altered flocculation and morphological phenotypes, which exhibit changes in chromosomal size. Genome resequencing suggested that torula yeast is triploid with six chromosomes and the mutants have multiple rearrangements including translocations having the TaqI recognition sequence at the break points. Thus, TAQing2.0 is expected as a useful method to obtain various mutants with altered phenotypes without introducing foreign DNA into asexual industrial microorganisms.

[1] Mitsubishi Corporation Life Sciences Limited, Tokyo Takarazuka Building 14F., 1-1-3 Yurakucho, Chiyoda-ku, Tokyo 100-0006, Japan. [2] Department of Life Sciences, Graduate School of Arts & Sciences, The University of Tokyo, Komaba 3-8-1, Meguro-ku, Tokyo 153-8902, Japan. [3] The Universal Biology Institute of The University of Tokyo, Hongo 7-3-1, Tokyo 113-0033, Japan. [4]These authors contributed equally: Taishi Yasukawa, Arisa H. Oda, Takahiro Nakamura. ✉email: kohta-pub2@bio.c.u-tokyo.ac.jp

It is important to maximize the use of biological functions and the efficiency of production of useful substances. Biotechnology enables microorganisms to improve their ability to produce substances and to synthesize new ones, particularly through genome improvement technologies such as genome editing, which have recently seen remarkable advancement[1,2].

Genome editing technologies such as CRISPR/Cas9 are very effective when the number of target genes to be modified is small, and responsible genes have already been identified. Meanwhile, in quantitative phenotypes represented by many biological functions, a large number of unidentified genes form a network to control their phenotypes in a complex manner. Random mutation technologies that generate a broader range of genomic diversity are effective in improving such complex quantitative phenotypes[3]. Conventional methods for inducing mutations with drugs or radiation are typical examples, but the supply of useful gain-of-function mutations is gradually becoming saturated, because the same techniques have been utilized for a long time. The conventional methods used in mutagenesis experiments often cause a large number of meaningless point mutations, and it is thus sometimes difficult to identify causative genes. In addition, there are issues to be overcome, such as the need for time-consuming processes (e.g., strain purification by backcrossing), to establish useful and phenotypically-stable strains.

An approach to overcoming these issues would be a technology that randomly induces large-scale genomic rearrangements, including copy number variations (CNVs) and translocations (TLs), while inducing single nucleotide variations (SNVs) to a lesser extent. One cutting-edge example[4–6] is the synthetic chromosome rearrangements and modifications by *loxP*-mediated evolution (SCRaMbLE) system, a core technology of the synthetic yeast genome project, Sc2.0 (Sc is an abbreviation for the budding yeast *Saccharomyces cerevisiae*)[7]. This technology utilizes haploid budding yeast with the genome replaced by a fully synthetic copy. A *loxPsym* site, which is the target sequence of the site-specific recombinant enzyme Cre recombinase, is inserted into the 3′ untranslated region of each of ~5000 nonessential genes. Chromosomal rearrangements, such as deletions, duplications, and translocations of genes between *loxP* sequences, are caused by induction of the Cre enzyme introduced by plasmid, thereby enabling the creation of a mutant pool with a diverse genetic constitution and characteristics.

To date, mutants resistant to alkali environments[8], heat, ethanol, and acetic acid[9] have been obtained, and the productivity of violacein, β-carotene[10,11], prodeoxyviolacein[12], carotenoids[13], and betulinic acid[14] have been improved by alteration of metabolic pathways. In addition, the establishment of biosynthetic systems for violacein and penicillin using xylose as a carbon source has been reported[15]. However, the SCRaMbLE system requires that an artificial chromosome with a *loxP* sequence be pre-constructed in the cells. Although there have been significant technological advances in genomic synthesis, it still remains time-, cost-, and labor-consuming, and obtaining artificial cells for the SCRaMbLE system in other organisms is not easy. The SCRaMbLE system has fundamental challenges, such as the rearrangement site being limited to the 3′ region of nonessential genes, and hence a more general-purpose technology has been awaited.

We have recently developed a large-scale genome restructuring technology called the TAQing system, which simultaneously induces multiple random DNA double-strand breaks (DSBs) and rejoins them, in the genomes of budding yeast and *Arabidopsis thaliana*, thereby improving phenotypes rapidly through genome restructuring[16]. In the original TAQing technology, a restriction enzyme called TaqI, which is derived from *Thermus thermophilus* HB8 and recognizes the 4-base sequence TCGA, is transfected into cells and expressed under the control of an inducible promoter for transient expression. Then, the cells are gently warmed to partially and temporarily activate TaqI, which randomly induces DSBs at TCGA sites throughout the genome, regardless of translated or untranslated regions, and then genomic rearrangements via DNA repair. This method enables the induction of large-scale genomic rearrangements in a variety of species without creating cells with artificial genomes that incorporate *loxP* sequences in advance. This method can be applied to plants whose genome size is 10 times larger than that of budding yeast.

The TAQing system would be effective even in many nonconventional yeasts, such as brewer's yeast, sake yeast, and industrial yeast (*Candida utilis*), which are imperfect fungi that do not undergo normal meiosis and are thereby impossible to be improved by conventional crossbreeding. If the TAQing system can be applied to microorganisms of high industrial utility while avoiding these disadvantages, it is considered as a powerful tool for artificial genome evolution technology that is expected to complement genome editing and SCRaMbLE technology.

Another industrial problem is the social acceptance of genetically modified organisms. General genome editing and SCRaMbLE technology are normally premised on the intracellular transfection of foreign DNA fragment sequences derived from heterologous species, and their application in foods, etc. may prompt new social discussions in the future[17,18]. A newly generated modified organism is subject to recombinant DNA regulations if it contains incoming DNA, and some industries are required to avoid these regulations. To overcome this problem, Serero et al. reported the transformation-free meiotic reversion system for yeasts to induce recombination-dependent genome diversification of sterile polyploid yeasts[19].

Recently, protein transfection technology[20,21] has begun to be used to introduce recombinant enzymes such as CRISPR/Cas9, Cre/*loxP*, and TALEN, into cells as proteins without introducing foreign DNA. Protein transfection includes invasive physical delivery methods and minimally invasive biochemical delivery methods. The former includes electroporation[18], plasma discharge[22], and the particle gun method[23]. In these methods, a complex consisting of CRISPR/Cas9 and guide-RNA, or a TALEN protein, can be delivered directly into live cells without DNA to induce its function. Examples include mammalian cell injury treatments[24,25], plant selective breeding[18,22,23,26–28], and breeding as microbial cell factories[29,30]. Interestingly, it has also been found that the delivery of genome editing enzymes as proteins into cells is associated with fewer editorial errors than delivery using DNA vectors[17,25]. However, these physical delivery methods are generally associated with high cell mortality and require special equipment/facilities and careful consideration of the conditions.

Minimally invasive delivery methods include those using biochemical carriers[31–33] or inactivated viruses[34,35]. These methods are operationally complicated, e.g., it is necessary to covalently link a carrier and target protein through an appropriate spacer in advance. In recent years, as a method to eliminate this bottleneck, a carrier that is easily delivered into cells by simply mixing it with a protein solution has been developed. Lipid-[36], polymer-[21], or cell-penetrating peptide (CPP)-based carriers such as Xfect system[37] have been reported, and all have low cytotoxicity and high delivery efficiency. Ideally, proteins should not be readily degraded until they are delivered to the target organelle, and a method using a CPP (e.g., the Xfect carrier) is thus preferred from this point of view. An Xfect/protein complex is internalized into cells via caveola-dependent endocytosis and macropinocytosis, and the rate of degradation is slow, since the proportion of colocalization with endosomes/lysosomes is only 10%[38]. It is thus considered optimal for intracellular transfection of an enzyme for genomic rearrangement.

We then aimed to solve the following two challenges simultaneously by combining the CPP method and the TAQing system: (i) Elimination of off-target mutations associated with heterologous DNA transfection and (ii) Modification of quantitative phenotypes largely through large-scale genomic rearrangements, while inducing only a few point mutations. This enables the easier discovery of unutilized genetic resources/potential related to fermentation properties, substance production capacity, etc. in various species such as brewing microorganisms.

In this study, we used C. utilis[39] (Cu), which has been widely used in industrial fields and is generally recognized as safe. Consequently, we have achieved the following: First, we established a minimally invasive method for transfecting any foreign active protein in a nucleic-acid-free and highly efficient manner using the CPP method for 2 yeast strains, S. cerevisiae (Sc) and Cu. Second, we developed a TAQing system of the protein transfection type (TAQing2.0) by directly delivering TaqI into Cu using the above method. Third, for Cu mutants with altered aggregation phenotypes obtained by the TAQing2.0 system, long- and short-read resequencing was performed to confirm the occurrence of large-scale chromosomal rearrangements at multiple sites.

Strains obtained by the TAQing2.0 system can be handled in the same manner as naturally occurring mutants because no foreign DNA has been introduced, and hence this method can be applied to fungi or other species that are incapable of sexual reproduction and of which it is difficult to obtain variants by crossbreeding. The TAQing2.0 system is thus considered extremely versatile and practical, and it is expected to make a tremendous contribution to the future development of biotechnology.

## Results

**Introduction of exogenous proteins into living cells by the CPP method.** Methods for introducing proteins, enzymes, and peptides into live cells using CPP-based protein transfection reagents have been developed primarily for animal cells, which lack cell walls. Meanwhile, a protein has also been reported to be successfully transfected into plants[27]. Our goal is to establish a method to induce genome rearrangements by transfecting endonuclease proteins into non-conventional yeasts such as Cu. However, to our knowledge, similar attempts in yeast and fungi have not been reported. We thus attempted first to introduce a foreign protein by the CPP method after spheroplasting by removing the cell wall of Sc and then further to find optimal conditions for Cu. An experiment was conducted to introduce β-galactosidase (β-Gal) into cells using spheroplasts prepared from the Sc strain S288c and the Cu strain NBRC0988. Intracellular transfection was quantitatively analyzed by the activity of β-Gal in cell extracts. A method (Xfect method) using Xfect, a transfection reagent, was employed[38]. Briefly, β-Gal and Xfect were pre-mixed to form a complex, which was then incubated with yeast cells.

In both yeasts, the β-Gal activity of cell extracts after removal of cell debris was increased when spheroplasts were incubated with Xfect (+) and β-Gal. More importantly, the live cells with intact cell walls treated with Xfect (+) and β-Gal were also found to show higher activity than those without Xfect (−) (Fig. 1a). In addition, the Xfect-treated cells of both yeasts exhibited similar levels of β-Gal activity irrespective of the spheroplasting. We did not observe any severe toxicity of the Xfect reagent for both yeast species (Supplementary Fig. 1). Notably, the β-Gal activity in the cell extract prepared from intact cells was generally higher in Cu than in Sc, indicating that protein transfection by the CPP method is more efficient in Cu.

These experiments alone cannot exclude the contribution of proteins adsorbed to the cell membrane. Following the introduction of β-Gal into cells by the CPP method, the cell surface was treated with trypsin to digest and remove proteins adsorbed to the cell wall. The β-Gal activity was then measured to assess the amount of intracellular protein uptake[32,40]. The results showed that a sufficient amount of β-Gal activity was detected, even in cells treated with trypsin, indicating that β-Gal was certainly transfected into cells (Fig. 1b). Green fluorescent protein (GFP), which is fused to a membrane-penetrating peptide, has been reportedly transfected in live cells of C. albicans via endocytosis[32]. Endocytosis is considered to be the primary mode of action for intracellular uptake of foreign proteins or peptides using Xfect as a carrier[38]. These results led us to propose a minimally invasive method of protein transfection into yeast cells with intact cell walls by Xfect method.

The intracellular transfection of β-Gal was then detected using live-cell imaging to support the above colorimetric results. The β-Gal/Xfect complexes were mixed with live cells of Sc or Cu, excess complexes were removed by washing, and then SPiDER-βGal, a fluorescent reagent, was added. SPiDER-βGal has cell membrane permeability, is converted to a quinone methide intermediate by an enzyme–substrate reaction with β-Gal, and exhibits fluorescence and intracellular retention by covalent binding with SH groups of nearby proteins[41]. Fluorescence signals were consequently noted inside cells and around the periphery of cells after addition of Xfect (+) for both Sc and Cu (Fig. 1c). In addition, almost no fluorescence signals were detected when no Xfect (−) or SPiDER-βGal (−) was added (Supplementary Fig. 2). These results demonstrated that the fluorescence image detected was not a false-positive, such as autofluorescence derived from cell stress associated with Xfect treatment.

Since a higher fluorescence intensity was observed in the cell periphery in Fig. 1c, the possibility that the β-Gal/Xfect complexes adhere to the cell membrane and react with SPiDER-βGal on the membrane to become fluorescent could not be ruled out. The cells were thus reacted with the β-Gal/Xfect complexes and suspended in citrate buffer (pH 4.0) after washing to inactivate β-Gal on the cell membrane. The citrate buffer was immediately replaced with normal phosphate-buffered saline (PBS) (pH 7.4), followed by addition of SPiDER-βGal for cell observation. Strong fluorescence signals at the cell periphery were attenuated, but intracellular fluorescence signals were still detected at a high level (Fig. 1c). These results demonstrated that fluorescence signals detected in the yeast cells were not derived from β-Gal adsorbed on the cell membrane but were attributable to intracellularly transfected β-Gal. The Xfect was thus found to function as a protein carrier and β-Gal to be intracellularly delivered while maintaining its activity.

Next, the differences between Sc and Cu were assessed in terms of the intracellular delivery effects of Xfect. The concentration of SPiDER-βGal was 0.2 μM for Cu, compared with 1.0 μM for Sc, as shown in Fig. 1c. This is because the fluorescence signals are saturated following addition of 1.0 μM of SPiDER-βGal to Cu under the same excitation laser intensity and exposure time conditions as for Sc. The intracellular fluorescence level was significantly higher in Cu than in Sc (Fig. 1d) when the level was measured quantitatively in the experiment, Xfect/citrate-treated cell images, shown in Fig. 1c. The β-Gal/Xfect complexes are thus seemingly more efficiently incorporated in Cu cells than in Sc cells. This speculation is consistent with the results of the colorimetric method shown in Fig. 1a, b.

We have thus established a method in which proteins/enzymes can be easily transfected into yeast cells while maintaining their activity even in Cu, a non-laboratory type yeast with a strong chitin cell wall.

**Preferable conditions for protein transfection into Cu by the Xfect method.** The transfection efficiency of a foreign protein is

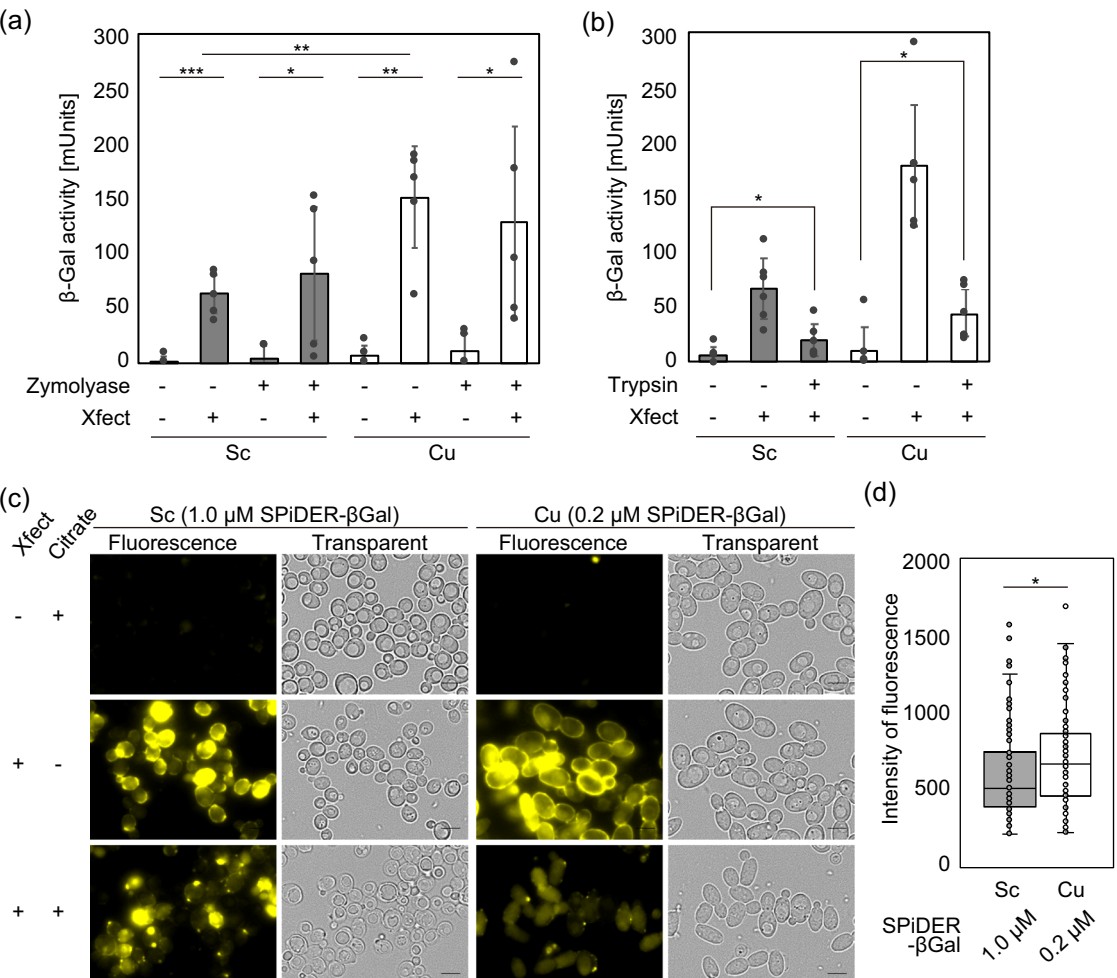

**Fig. 1 An Xfect-based protein transfection method can introduce β-Galactosidase (β-Gal) into budding yeast cells. a** Effects of spheroplastization on protein transfection. *Saccharomyces cerevisiae* S288c (Sc) and *Candida utilis* NBRC0988 (Cu) cells (Zymolyase −), or their spheroplasts (Zymolyase +), were incubated with β-Gal in the presence (+) or absence (−) of Xfect™ Protein Transfection Reagent (Takara Bio Inc., Shiga, Japan), and the β-Gal activity was measured by a colorimetric method. Bars and error bars, respectively, represent the mean and standard deviation from five independent experiments ($n = 5$) and one-tailed Welch's $t$-test was applied (*$p < 0.05$, **$p < 0.01$, ***$p < 0.001$). Experiments were performed in triplicates. **b** Effects of trypsin digestion of membrane-absorbed β-Gal on β-Gal quantification. β-Gal was introduced to Sc and Cu cells as in (**a**), which were subsequently treated with (trypsin +) or without (trypsin −) 1% trypsin in phosphate-buffered saline (PBS, pH 7.4) in order to digest membrane-absorbed β-Gal and β-Gal/Xfect complexes. The activity of intracellular β-Gal was determined as in (**a**). Bars and error bars, respectively, represent the mean and standard deviation from six independent experiments ($n = 6$) and one-tailed Welch's $t$-test was applied (*$p < 0.05$). Experiments were performed in triplicates. **c** Fluorescent-based detection of β-Gal activities inside Sc and Cu living cells. β-Gal- (Xfect −) or β-Gal/Xfect-mixed (Xfect +) cells, with (citrate +) or without (citrate −) subsequent β-Gal inactivation by citrate buffer (pH 4.0), were incubated with SPiDER-βGal of the indicated concentration, and their images were acquired using the fluorescence microscopy BZ-X700. The scale bar is 5 μm, and a representative image for each experiment is shown. **d** Quantitative comparison of intracellular β-Gal activity between Sc and Cu. Variability in the intracellular fluorescence signals between Sc and Cu cells were determined using citrate-treated cell images (**c**) and Hybrid cell-counting tool. The data of Sc (1.0 μM SPiDER-βGal) and Cu (0.2 μM SPiDER-βGal) are shown as the box plots ($n = 120$, three fields per replicate). The center line is the median, bounds are the 25th and 75th percentiles, and whiskers are ±1.5 IQR. Error bars indicate SD from cell counts. Asterisk indicate significant differences analyzed using one-tailed Welch's $t$-test, *$p < 0.05$.

an important factor in the delivery of an active protein to the cellular target organelle at an appropriate timing and the continuous expression of its physiological function. We therefore searched preferable conditions for protein transfection by altering buffer reagents. The use of HEPES, a Good's buffer, has been reported to increase the efficiency of protein transfection into mammalian cells[42]. Thus, the optimal conditions for transfection into Cu were investigated on the basis of the intracellular activity of β-Gal delivered. According to a common protocol for animal cells, Xfect should be used at 60 to 80% cell confluence. For budding yeast in the above section, the Xfect method was applied to cells in the exponential (log) phase of growth in yeast-peptone-dextrose (YPD) liquid medium. We then compared three types of

media, YPD, synthetic defined (SD), and yeast extract with supplements (YES). The highest β-Gal activity was observed in YPD medium (Supplementary Fig. 3a). Next, the efficiency of the Xfect method was assessed at each stage of proliferation in YPD medium in the early exponential phase ($OD_{600}$ value up to 1), middle exponential phase (value 1–10), late exponential phase (value 10–20), and stationary phase (value 20–23), and the highest β-Gal activity was detected when cells in the early exponential phase (value up to 1) were used (Supplementary Fig. 3b).

We furthermore examined the optimal conditions for the transfection of β-Gal/Xfect into Cu cells. In the standard procedure, cells are suspended in PBS and incubated at 37 °C

for 60 min. When various exposure times and treatment temperatures were tested, we found 30 to 120 min and 20 to 37 °C were appropriate, respectively (Supplementary Fig. 3c, d). We also examined the composition of the buffer used to suspend cells (Supplementary Fig. 3e–g); the highest β-Gal activity was observed with salt-free 2-morpholinoethanesulphonic acid (MES) (pH 6.0) (Supplementary Fig. 3e, f). We note that this result is likely to reflect an optimal condition for protein transfection, rather than for β-Gal activity, since β-Gal had similar activity between pH 5.0 and pH 10.0, regardless of the buffer type (Supplementary Fig. 3g). Thus, the most important factor in the optimization was the extracellular environment at the time of protein transfection (Supplementary Fig. 3f).

We finally established an optimization procedure by combining each condition that showed the maximum β-Gal transformation efficiency, that is a condition in which Cu is incubated in YPD medium until the early exponential growth phase and brought into direct exposure to the β-Gal/Xfect complexes in the salt-free MES (pH 6.0). The efficiency of β-Gal transfection by the original procedure was 231 mUnits while 607 mUnits in the optimized one, indicating that the latter had 2.6-fold higher efficiency (Fig. 2a). Such improved effects by the optimized protocol were again confirmed by the SPiDER-based measurements of the intracellular β-Gal activity (Fig. 2b, c).

We further assessed whether the optimized protocol were also effective for proteins other than β-Gal. According to the quantitative comparison of Alexa Flour 488-conjugated goat IgG (Alexa488-IgG) with a mean molecular weight of 160 kDa, the transfection efficiency was higher in MES than in PBS (Fig. 2d, e). GFP was assessed in the same manner, and the transfection efficiency was found to be higher when MES was used as a cell suspension buffer in comparison with PBS (Fig. 2f, g). Furthermore, foreign proteins were transfected into Cu cells only when those proteins were incubated with Xfect (Fig. 2d, f). In conclusion, the optimization procedure of the Xfect method for Cu was applicable for various proteins.

**Genome breeding of Cu (torula yeast) using the protein-based TAQing system (TAQing2.0).** We have developed the TAQing system, which simultaneously induces multiple genomic DSBs via transient heat-activating/inducing of TaqI in Sc and *A. thaliana* cells[16]. The DNA cleavage activity of TaqI is maintained at a low level at room temperature but markedly activated at a higher temperature[16]. Since one TCGA sequence theoretically appears per 256 base pairs in the genome, breaks can be introduced at many genomic sites, and this allows for the induction of multisite chromosomal rearrangements by facilitating rejoining of the cleavage sites. In addition, we analyzed the genomic DNA sequence of Sc mutants obtained by the TAQing system, and found that SNVs were limited, but insertions-deletions (InDels), homologous/nonhomologous chromosome translocations (TLs), and CNVs occurred frequently. The recognition sequence (TCGA) was often detected at the break points in inter-chromosomal TLs. The TAQing system can efficiently generate mutant strains for quantitative phenotypes known to involve a large number of genes[16]. In fact, it has been shown that a variety of mutants with altered phenotypes can be obtained in a very short period, such as yeast with both xylose-assimilation ability and high-temperature fermentation ability and plants with various morphological changes[16]. The TAQing system thus enables rapid genome improvement even by skipping sexual reproduction processes including meiosis and sporulation.

We thus considered that application of the TAQing system to Cu, which has been widely used as edible/fodder yeast and whose safety has been approved by the United States Food and Drug

Administration[43], would overcome the challenges of industrial use that only limited mutations are obtained in classical breeding of this yeast, since Cu is a high-order polyploid non-conventional yeast that has lost their sporulation ability. We also considered that combination of the above CPP method and the TAQing system may solve the problem of transfecting foreign DNA into yeasts for edible/fodder use.

We thus designed another version of TAQing system using the CPP-mediated protein transfection to Cu. Since Cu reproduces asexually, chromosome-level recombination is limited to accidental events during mitosis and its frequency is considered extremely lower than homologous recombination during meiosis. Large-scale genomic rearrangements induced by the TAQing system are therefore expected to produce valuable mutants than ever obtained. We additionally expected to take advantage of the TAQing system, which induces genomic rearrangements more effectively in polyploid species[16], since the Cu genome is estimated to be polyploid[43,44]. From these, we proposed alternative derivative of the TAQing system based on the direct protein delivery into the cell nucleus (the TAQing2.0 system) (Fig. 3a).

The protein size that can pass through nuclear pores by diffusion is reportedly 9 nm in diameter or less than 60 kDa in molecular weight[45]. The estimated molecular weight of TaqI is ~31 kDa, which can easily clear the hurdle, though we fused TaqI with a nuclear localization signal (NLS: PKKKRKV) of the SV40 large T antigen at its N-terminus. This NLS-tag is also expected to minimize intracellular degradation and rapidly deliver/localize in the target cell nucleus.

NLS-TaqI with a 6×His-tag for purification was expressed with a pET system in *Escherichia coli*, and then NLS-TaqI was purified under high concentrations (3.3–4.7 mg/mL) in a cobalt resin affinity column (Supplementary Fig. 4a). The purified NLS-TaqI showed the same cleavage pattern as commercially available TaqαI (New England Biolabs Inc.), demonstrating that no changes occurred in the substrate specificity (Supplementary Fig. 4b).

Cell viability is a parameter for assessing the induction of DSBs by transfected NLS-TaqI. The viability of TaqI-transfected Sc cells after heat treatment has been reported to decrease to 20% or less in haploids and 40% in diploids[16]. The NLS-TaqI/Xfect complex was introduced into Cu cells using the optimized procedure described above. The cells were washed and incubated in YPD medium at 30 °C for 30 min for recovery. Subsequently, the Cu cells were incubated at 38 °C for 90 min to temporarily and partially activate TaqI. As controls, we employed samples incubated at 30 °C for 90 min or those transfected with nontoxic bovine serum albumin (BSA). The diluted cells were then plated on YPD agar medium, and the viable colonies were counted (Fig. 3b, c).

When the viability of cells treated with the Xfect reagent alone was designated as 100%, the viability score of cells incubated with the NLS-TaqI/Xfect complex decreased to 24.1% at 30 °C and markedly to 8.5% at 38 °C (Fig. 3c). Increasing the incubation temperature resulted in marked reduction of cell viability. Thus, the activity of transfected NLS-TaqI was indeed temperature dependent. Meanwhile, the viability of the BSA-treated control unchanged at much higher levels at 30 and 38 °C. Possibly, either the thermal resistance of cells was increased or the cytotoxicity of Xfect was reduced owing to incubation with BSA. These results suggested that NLS-TaqI was transfected into Cu cells and DSBs were induced in the genome.

Application of the TAQing system to Sc induces cellular morphological alterations[16]. Changes in chromosome size in these cells were often noted by pulsed-field gel electrophoresis (PFGE)[16]. For analysis of Cu after applying the

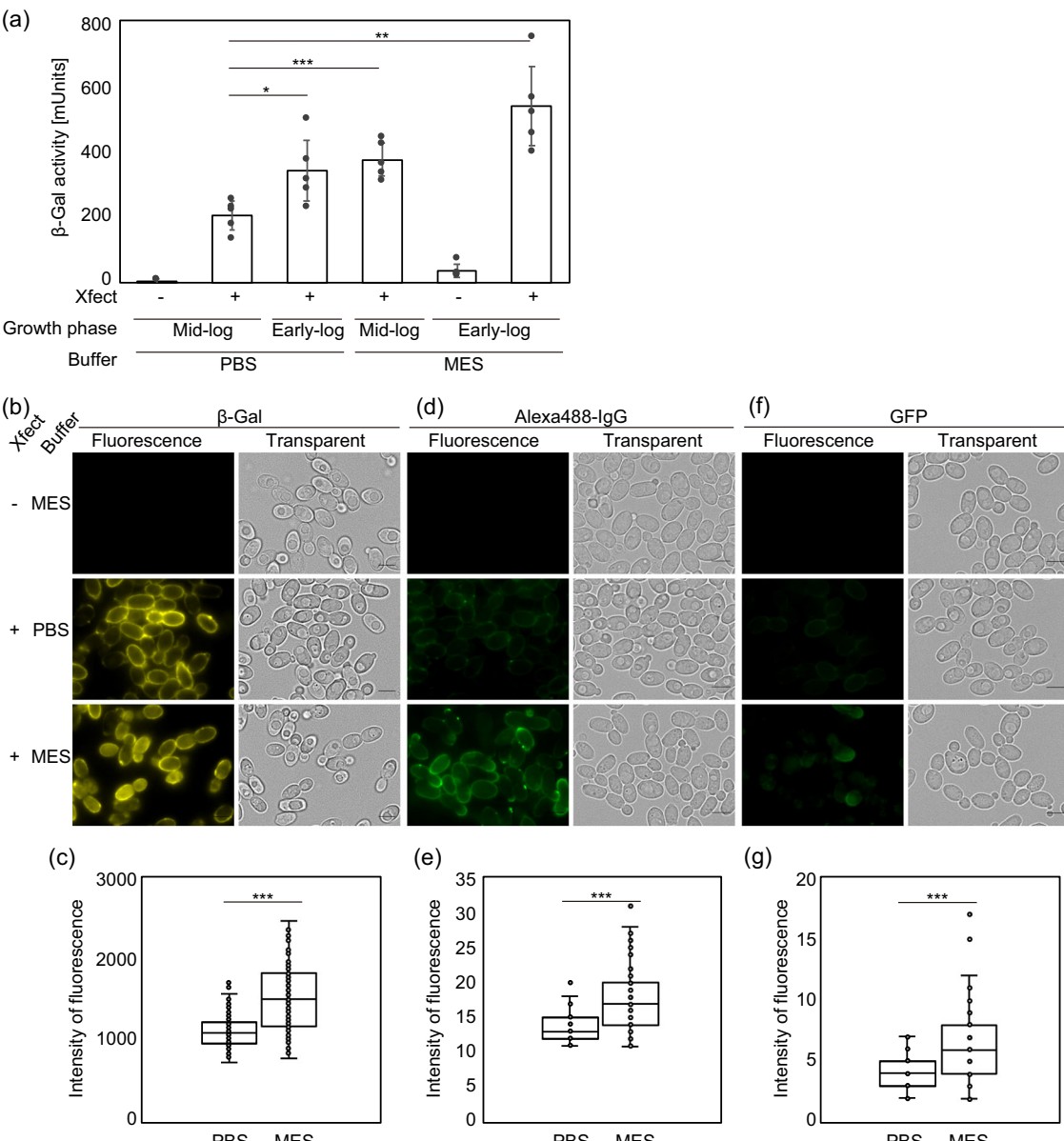

**Fig. 2 Optimizing the Xfect method for protein transfection into Cu. a** Effects of growth phase and buffer on transfection efficiency into Cu. Cu intact cells harvested at early-log phase (OD$_{600}$ < 1) or mid-log phase (OD$_{600}$ of 1–10) were suspended in PBS (pH 7.4) or in MES (pH 6.0) without salts, and subjected to protein transfection with Xfect. The total β-Gal activities were quantified using the colorimetric method. Bars and error bars, respectively, represent the mean and standard deviation from five independent experiments ($n = 5$) and one-tailed Welch's $t$-test was applied (*$p < 0.05$, ***$p < 0.001$). Experiments were performed in triplicates. **b–g** Visual assessment of PBS and MES on transfection efficiency into Cu. Cu cells in early-log phase were suspended in PBS (pH 7.4) or MES (pH 6.0) without salts, and transfected with β-Gal (**b** and **c**), Alexa488-IgG (**d** and **e**), or GFP (**f** and **g**). Mock experiments without Xfect, serving as negative control, were also performed. **b, d, f** Representative images are shown. The scale bars are 5 μm. Note that, in (**b**), β-Gal activity was visualized by 0.2 μM SPiDER-βGal. **c, e, g** Quantification results are shown by box plots. Quantification of fluorescence intensity in MES (pH 6.0) was compared with that in PBS (pH 7.4) by using acquired images (**b, d, f**) and the Hybrid cell-counting tool. The sample size of $n = 145$ (**c**), $n = 47$ (**e**), $n = 35$ (**g** PBS), or $n = 71$ (**g** MES) from 3 to 5 fields per replicate were analyzed. Asterisk indicate significant differences analyzed using one-tailed Welch's $t$-test, ***$p < 0.001$.

TAQing2.0 system (TAQed Cu), we picked up clones with smaller colony size followed by selecting strains with cellular morphological changes. Of 1352 colonies of TAQed Cu formed on YPD agar medium, 38 small colonies were observed under a bright-field microscope. Six of these strains had changes in their flocculation phenotype. They were consecutively subcloned 5 times in YPD medium. After the passages, 2 strains still maintained their aggregability (Fig. 3d). We further analyzed these 2 strains (AG4 and AG9) exhibiting stable strong aggregability after the passages. RNA-seq experiments for AG4

and AG9 strains supported their hyper-flocculation phenotypes, since we observed marked increase in the expression of flocculation genes such as *CuFLO1* and *CuFLO5* in these strains (Supplementary Fig. 5 and Supplementary Table 1a).

Changes in chromosome size were analyzed by PFGE for Cu wild type, AG4 and AG9 strains: Cu wild type (WT), AG4, 5-round passaged AG4 (AG4×5), AG9, and 5-round passaged AG9 (AG9×5). Both AG4 and AG9 had altered chromosome sizes (Fig. 3e and Supplementary Fig. 6). No changes were detected in the band sizes before and after the 5-round passages, indicating

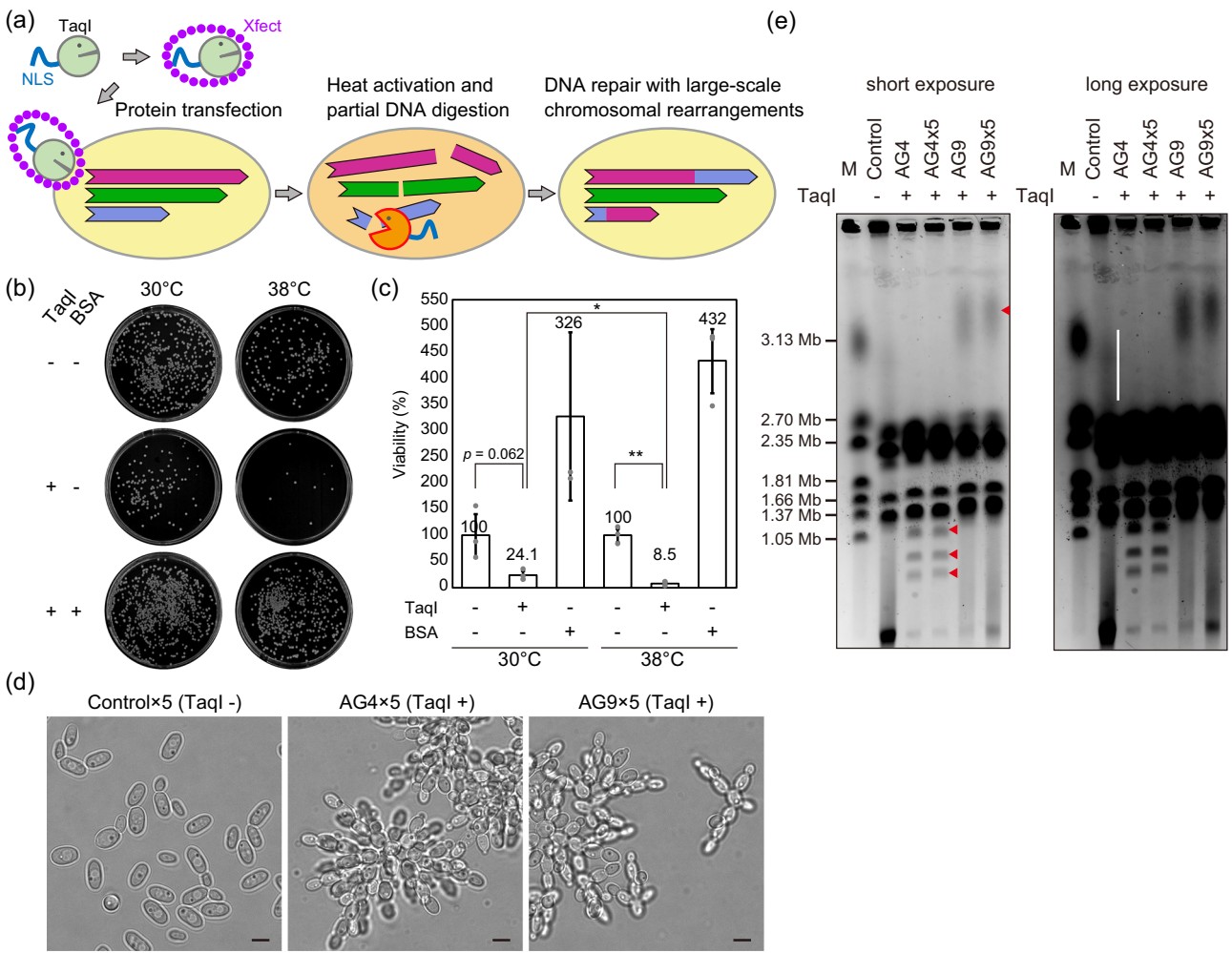

**Fig. 3 Application of the protein-based TAQing system, TAQing2.0, to the non-conventional yeast Cu. a** A schematic diagram describing a procedure of the TAQing2.0 system. **b**, **c** Reduction of cell viability by activation of TaqI (TAQed) introduced to Cu cells by the TAQing2.0 system. Cells subjected to the optimized TAQing2.0 with no protein, TaqI, or BSA were incubated at 30 or 38 °C for 90 min, and plated onto YPD plates. Plates were then incubated at 30 °C for a few days. **b** Representative images of plates (9 cm diameter) with formed colonies after the treatment. **c** Cell viabilities after the TAQing2.0 system application. Colony forming units (CFU) were calculated for each experimental group shown in (**b**), and cell viabilities were determined by setting CFU without protein (i.e., TaqI- BSA-) to 100%. Bars and error bars, respectively, represent the mean and the standard deviation from three independent experiments ($n = 3$) and one-tailed Welch's $t$-test was applied (*$p < 0.05$, **$p < 0.01$). Experiments were performed in triplicates. **d** Morphological images of wild type (WT) Cu×5 (Control after five passages), AG4×5 (TAQed mutant AG4 cells after five passages), and AG9×5 (TAQed mutant AG9 cells after five passages). Bright-field images were obtained by microscopy BZ-X700. The scale bars are 5 μm. **e** Chromosome sizes of WT Cu (Control) and TAQed Cu (AG4, AG4×5, AG9, and AG9×5) cells. Chromosomal DNA prepared from Cu cells along with size marker DNA fragments (Bio-Rad Catalog #170-3667) were analyzed by pulsed-field gel electrophoresis. Red triangles in the short exposure panel indicate appearance of bands in the TAQed mutants. We noticed that the longest chromosome in wild type (chromosome I) exhibited smear bands (the vertical bar in the long exposure panel) possibly due to heterogeneity of rDNA repeat number. Instead, we detected shortened chromosomes in AG4 and AG4×5 (the three red triangles). The AG9 and AG9x5 mutants had elongated chromosome I (the red triangle). Since a duplication at a position proximal to rDNA region was observed in AG9x5 (Fig. 4a), it is possible that this duplication may affect the rDNA stability.

that the genomic structure, once rearranged, was stably maintained at least over several rounds of passages. Figure 3e showed that two bands, indicated by leftwards red arrow, had been clearly shifted up in both AG9 and AG9×5, and this would be perhaps due to large-scale structural alterations around rDNA in ChrI, partial duplication, or aneuploidy in ChrIII (described in details below).

**Whole-genome resequencing of Cu mutants obtained by TAQing2.0.** We then analyzed the whole-genome sequences of AG4 and AG9. We first determined the reference genome of the original Cu strain to accurately identify TAQing-dependent rearrangements. The draft genome of the Cu WT was previously reported using a 454/Roche sequencer[46]. Rupp and coworkers reported that the ploidies of Cu and *Cyberlindnera jadini* (*C. jadinii*) were triploid and diploid, respectively, based on the single nucleotide variation (SNV) analysis of Cu NBRC0988 and its possible ancestor *C. jadinii* NRRL y-1542[44]. Meanwhile, the frequency of local SNPs in *C. jadinii* has been reported to be tri-ploid, tetraploid, or haploid (large deletion), instead of diploid in the specific region (several hundred kilobases) of each scaffold.

Since the genome composition of torula yeast remains controversial as described above, we employed a combination of long-/short-read sequencing to obtain a more accurate reference genome sequence (Supplementary Table 2). Thirty contigs (total length: 14,097,946 bases) were generated through

both PacBio Sequel and Illumina MiSeq data with the assembler MaSuRCA[47]. Meanwhile, when assembly was performed with the assembler FALCON, 24 primary contigs and 234 alternative haplotigs were obtained.

By the manual connection of the contigs derived from MaSuRCA and FALCON using BLAST[48] for local alignment search and YASS[49] for genomic similarity search, we assumed that torula yeast is triploid with six chromosomes (Fig. 4a and Supplementary Table 3). In consistent with this assumption, we detected 5–20 or more repetitive units of the telomeric sequence GGGTGTCT, which is similar to the telomeric repeat of Sc[50], at each end of these six chromosomes. Notably, we identified an independent short extra-chromosome with telomeric repeats at both ends, consisting of copy of ~700 kb segment of the right arm of chromosome-II (hereafter referred to as ChrII, Fig. 4b).

Genomic sequencing of AG4 and AG9 using Illumina NovaSeq generated data of 4–12 Gb (at least 280-fold coverage, the Cu genome size is 14 Mb). The sequence obtained was mapped to the reference genome to assess large-scale genomic rearrangements based on changes in the frequency and coverage of SNVs. We found that AG4 had two break-induced repairs (BIRs) in ChrI and ChrIV, nine gene conversion (GCV) in ChrI, ChrII and ChrIV, and three large deletion in ChrI and ChrII (Fig. 4b, c and Table 1). AG9 had one BIR in ChrIII, five GCV in ChrI and ChrIII, one large deletion (Del) over 67 kb in ChrIII and one smaller Del about 6 kb in ChrI (Fig. 4b and Table 1). Figure 4d illustrates a large Del in ChrIII and the large Del was found in TaqI-recognition sequence (TCGA), and also confirmed by PCR amplification (Fig. 4d). SNVs and InDels were detected at three sites in AG4 and four sites in AG9 (Table 1). These results showed that AG4 and AG9 had multiple large genomic rearrangements.

Next, to detect nonhomologous chromosomal recombination such as TLs, we focused on reads mapped between different contigs and extracted regions where the boundary sequence was TCGA, the recognition sequence of TaqI. In addition, the coverage of AG4 and AG9 was normalized to the WT one to quantitatively estimate chromosome copy numbers in the extracted regions. Consequently, we found AG4 has one nonhomologous TL (the actual coverage data are shown in Fig. 4a, magenta is AG4). A TL event was detected between ChrI and ChrIV, and their physical linkage was confirmed by PCR amplification (Fig. 4b, e). AG9 had aneuploidy, i.e., ChrIII was tetrasomy, and the right arm of ChrI had an additional copy (partial duplication).

These results showed that the TAQing2.0 system, which is based on direct protein delivery into the cell nucleus, induced TaqI-mediated large-scale genomic rearrangements in Cu and enabled the generation of mutants with an altered flocculation phenotype.

## Discussion

In this study, we upgraded the TAQing system (using plasmid vector transfection) to the TAQing2.0 system (using protein transfection), which can induce genomic rearrangements even in non-sporulating industrial yeasts without introducing any foreign DNA (Fig. 3a). This technology is characterized by delivering restriction endonuclease proteins directly into the cell nucleus by the CPP-mediated protein transfection method (Fig. 3b, c). We confirmed by PFGE and genome resequencing that TAQing2.0 led Cu to efficiently induce TaqI-dependent large-scale genomic rearrangements (Figs. 3e and 4a–e). This technology is expected to be used in a variety of industrial microorganisms producing useful substances or fermented foods.

Here we established a versatile and easy protein transfection method for budding and torula yeast using CPP. The results

indicate that proteins such as β-Gal, antibodies, GFP, and endonucleases can be delivered into living cells while keeping their activities (Figs. 1 and 2). In comparison with other existing methods for protein transfection into various species having cell wall such as electroporation or membrane-penetrating signal peptide fusion[18,22,23,27–33], the CPP-mediated method (Xfect method) is very simple, only needs brief incubation of CPP with proteins, and does not require a fusion of signal peptide sequences or cell wall removal (Figs. 1 and 2b–g). In addition, the CPP-mediated method is minimally invasive and less toxic to both cell viability and protein activity compared to electroporation-based transfection.

In this study, we optimized the transfection conditions for Sc and Cu (Supplementary Fig. 3). The optimized conditions work more efficiently in Cu than in Sc (Figs. 1a and 2a). Two factors were turned out to be important: the employment of a MES buffer (pH 6.0) containing no salt (Supplementary Fig. 3e, f) and the usage of cells in the early exponential growth phase (Supplementary Fig. 3b). Zwitterionic good buffers such as MES and HEPES may contribute to lower the diffusion coefficient by neutralizing charge of Xfect conjugated foreign proteins with zwitterions, thereby facilitating cellular internalization of the protein, as suggested by Chen et al.[42]. The latter factor of growth phase may influence the cell wall porosity and the efficiency of biomolecule penetration due to the cell-cycle dependence of the molecular composition of the cell wall[51,52]. It should be also noted that the optimal conditions of the CPP method may be different from one yeast species to another. Therefore, one should optimize conditions for the CPP method when this technology is applied to other fungal organisms.

Since the genome structure of torula yeast has been rather controversial even in terms of ploidy and chromosome composition (13 chromosome scaffolds in computing analysis in Cu genomics)[43,44], we generated a more accurate reference genome sequence of the wild type Cu strain (Cu NBRC0988) by combination of long- and short-read sequencing. We finally estimated that Cu NBRC0988 is triploid with six chromosomes (Fig. 4a, b) including a short chromosome consisting of extra-copy (~700 kb) of the right arm of ChrII. ChrI is the longest chromosome harboring rDNA repeat (Supplementary Table 3). The PFGE profile of Fig. 3e also supported the model of six chromosomes. Detected bands were indeed around seven or so, the number of which is definitely smaller than that of the previous 13 chromosome model[43,44]. We also detected putative telomeric repeats at both ends of six chromosomes and the extra short chromosome. We uploaded this reference sequence in the genome database for the public use (accession number AP024664-AP024669).

We then assess the efficiency of the TAQing2.0 system in the generation of new mutants. In this study, two Cu mutants with altered flocculation phenotypes were obtained from 38 smaller colonies by the TAQing2.0 system (Fig. 3a–e). This frequency is 5.3% (2/38), which is much higher than the neglectably low frequency of natural occurrence of mutants with similar phenotype. In the original TAQing system, 11 isolates of 178 smaller Sc colonies (6.2%, 11/178) exhibited the hyper-flocculation or hypo-flocculation phenotypes[16]. Therefore, the TAQing2.0 system seems to be similarly effective in generating mutants with altered flocculation phenotypes to the original TAQing system.

The TAQing2.0 system is expected to efficiently generate mutants with various altered phenotypes other than those of flocculation and cell morphology, including quantitative traits controlled by a complex gene network favorable for some industrial purposes. Since the TAQing2.0 system makes it easier than the conventional breeding methods to determine the causative genetic alterations responsible for the relevant traits, we would be able to design in the future a strain with the synthetic

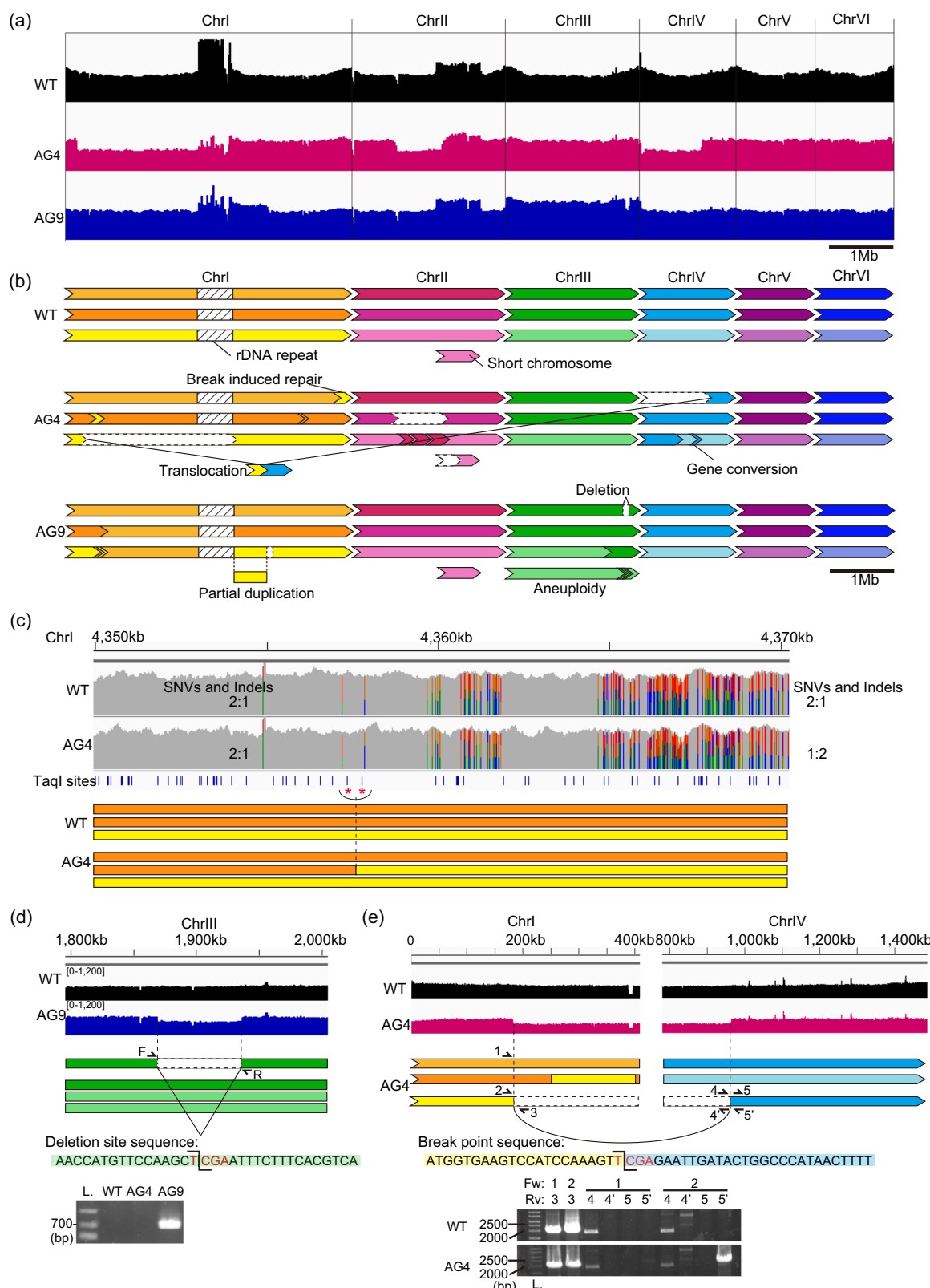

genome which harbors minimal sets of causative genes. It may be also possible to enhance the useful traits by repeatedly conducting the TAQing2.0-mutagenesis/screening cycle.

Here we demonstrated potential capability of the TAQing2.0 system, which can generate mutants of various asexual yeasts such as brewer's yeast, such as sake and beer yeasts without introducing any exogenous DNAs (see the phylogenetic tree in Supplementary Fig. 7). This may lower the hurdle of sporulation-based breeding of these organisms and relieve the difficulties of public acceptance genetically modified organisms carrying exogenous DNA sequences. The TAQing2.0 system could be applied to other filamentous fungi, such as *Aspergillus oryzae* for food fermentation and *Penicillium noctum* for antibiotics production. We also reported the extended TAQing

**Fig. 4 Chromosome structures and rearrangements in TAQed strains. a** Genome resequencing data of Cu WT (black), TAQed strain AG4×5 (magenta) and AG9×5 (blue) at the whole-genome view. Aligned NovaSeq reads were visualized by Interactive Genome Viewer (IGV). **b** Schematic diagrams of rearranged chromosomes in the TAQed strains. Within WT triploid chromosome sets, two alleles tend to have similar SNV and InDel patterns, which were shown in the same color. Genome rearrangements such as break-induced repairs, gene conversions, large deletions, a translocation, an aneuploidy were shown. **c** An example of break-induced repair. The upper panel displays the local view of aligned sequences of WT and AG4 around ChrI 4357 kb with their SNV and InDel patterns. The blue lines show the position of TaqI recognition sites in the WT chromosomes. The two TaqI sites with red stars are the potential homologous recombination locus as the proportion of SNV and InDel changes. The bottom panel shows schematic image of rearranged alleles. **d** An example of large deletion at the ChrIII in AG9 strain. Aligned reads, schematic image, and around the deletion locus are shown. The break point sequence matched the TaqI recognition sequence TCGA. The deletion was also confirmed by PCR. The primer positions were shown in the schematic image. **e** The local view of Translocation at the ChrI and ChrIV in AG4 strain.

**Table 1 Large-scale rearrangements, SNVs and InDels detected in TAQed mutant strains AG4 and AG9.**

| Mutant ID | Start position | | End position | | Type of rearrangement | Note |
|---|---|---|---|---|---|---|
| AG4 | ChrI | 183,236 | ChrIV | 964,883 | Translocation | |
| AG4 | ChrI | 183,237 | ChrI | 2,609,500* | Deletion | *Around rDNA |
| AG4 | ChrII | 725,780 | ChrII | 1,407,626 | Deletion | |
| AG4 | ChrII | 1,305,106 | ChrII | 1,546,790 | Deletion | |
| AG9 | ChrI | 3,209,917 | ChrI | 3,216,299 | Deletion | |
| AG9 | ChrIII | 1,869,226 | ChrIII | 1,936,358 | Deletion | |
| AG9 | ChrIII | | | | Aneuploidy | Whole chromosome III |
| AG9 | ChrI | 2,613,000* | ChrI | 3,209,916 | Partial duplication | *Around rDNA |
| AG4 | ChrI | 4,357,880 | ChrI | 4,541,153 | Break induced repair | |
| AG4 | ChrIV | 1 | ChrIV | 463,762 | Break induced repair | |
| AG9 | ChrIII | 1,705,647 | ChrIII | 2,119,607 | Break induced repair | |
| AG4 | ChrI | 249,379 | ChrI | 398,066 | Gene conversion | |
| AG4 | ChrI | 3,814,598 | ChrI | 3,819,195 | Gene conversion | |
| AG4 | ChrII | 751,384 | ChrII | 920,699 | Gene conversion | |
| AG4 | ChrII | 938,020 | ChrII | 1,317,635 | Gene conversion | |
| AG4 | ChrII | 1,318,690 | ChrII | 1,415,237 | Gene conversion | |
| AG4 | ChrII | 1,567,821 | ChrII | 1,572,073 | Gene conversion | |
| AG4 | ChrII | 1,581,636 | ChrII | 1,583,219 | Gene conversion | |
| AG4 | ChrIV | 671,337 | ChrIV | 673,840 | Gene conversion | |
| AG4 | ChrIV | 730,060 | ChrIV | 731,830 | Gene conversion | |
| AG9 | ChrI | 408,842 | ChrI | 409,294 | Gene conversion | |
| AG9 | ChrI | 409,925 | ChrI | 2,084,873* | Gene conversion | *Around rDNA |
| AG9 | ChrI | 409,925 | ChrI | 2,084,873* | Gene conversion | *Around rDNA |
| AG9 | ChrIII | 1,920,323 | ChrIII | 1,933,957 | Gene conversion | |
| AG9 | ChrIII | 1,934,656 | ChrIII | 1,936,316 | Gene conversion | |
| AG4 | ChrII | 681,967 | | | SNV | G>T |
| AG4 | ChrIII | 1,600,622 | | | SNV | ACTT>GTA |
| AG9 | ChrIII | 1,496,570 | | | SNV | G>A |
| AG9 | ChrIV | 1,409,750 | | | SNV | G>A |
| AG4 | ChrIV | 904,210 | | | InDel | G>GA,GC |
| AG9 | ChrIII | 1,496,562 | ChrIII | 1,496,566 | InDel | GCCTC>G |
| AG9 | ChrIII | 1,496,572 | | | InDel | T>TATA |

Precise mutation points could not be analyzed around repetitive rDNA sequences which are marked with *.

system (Ex-TAQing) using restriction enzymes other than TaqI[53]. The TAQing2.0 system can be applied to a wider range of organisms and phenotypes in combination with the Ex-TAQing system. By using these TAQing systems in a complementary manner with SCRaMbLE and genome editing technologies, we expect to be able to establish the next stage of modern genome breeding technologies.

## Methods

**Yeast strains and cultures**. *Candida utilis* NBRC0988 (ATCC 9950) was used as non-conventional yeast strains, and *S. cerevisiae* S288c was used for a comparative study. The former was obtained from the National Institute of Technology and Evaluation (Tokyo, Japan), and the latter from Summit Pharmaceuticals International Corporation (Tokyo, Japan). TAQed *C. utilis* mutants (AG4 and AG9) were generated by TaqI activation (TAQing2.0) as described below. Yeast cells were aerobically cultured at 30 °C in YPD, SD, or YES (30 g glucose, 5 g yeast extract, and 0.05 g each of Adenine, Uracil, Histidine, Leucine, and Lysine for 1 L). Spheroplasts were prepared by Zymolyase-100T (Seikagaku Biobusiness Corp.,

Tokyo, Japan) treatment in Tris-EDTA (TE) buffer (50 mM Tris-HCl, 150 mM NaCl, 5 mM EDTA, and 5% PEG6000)[54].

**Protein transfection into yeast cells by the Xfect™ reagent**. Proteins were transfected into yeast cells using the Xfect™ Protein Transfection Reagent (Takara Bio Inc., Shiga, Japan), according to the Manufacturer's instruction with modifications. Briefly, yeast cells ($4.0 \times 10^7$ cells/ml) suspended in PBS (pH 7.4) or 10 mM MES (pH 6.0) were mixed with the Xfect reaction buffer along with proteins, preincubated at room temperature for 30 min, and then washed with PBS to remove the reaction reagent. Proteins used were 4 μg β-Gal (accessory in Xfect reagent), 46 μg recombinant GFP (Abcam, Cambridge, CB2 0AX, UK), 5 μg Alexa Fluor 488 goat anti-rabbit IgG (Cell Signaling Technology, Danvers, Massachusetts, USA), and 50–70 μg recombinant NLS-TaqI (prepared in this study; see below). In experiments shown in Fig. 1b, protein/Xfect complexes absorbed on the extracellular membrane or cell wall were digested by 1% trypsin/PBS treatment at 37 °C for 5 min[32,40].

**Colorimetric β-galactosidase detection assay**. The β-Galactosidase (β-Gal) activity introduced into yeast cells was quantified by a previously-reported

colorimetric method to monitor o-nitrophenol that was produced from o-nitrophenyl-β-D-galactopyranoside (ONPG) by the enzyme[55,56]. Yeast cells suspended in Z-buffer (60 mM di-sodium phosphate, 40 mM sodium di-phosphate, 10 mM KCl, 1 mM magnesium sulfate, and 50 mM β-mercaptoethanol (pH 7.0)) were lysed by addition of chloroform and 0.1% sodium dodecyl sulfate (SDS), and the lysate was incubated with ONPG until the reaction was terminated by sodium carbonate. The lysate, after centrifugation to remove cell debris, was measured for $OD_{420}$ to calculate the OD value in the following formula.

$$OD = OD_{420}/(T \times V \times OD_{600} \text{ unit}):$$

$OD_{420}$: $OD_{420}$ of supernatant after the ONPG hydrolysis; $OD_{600}$: $OD_{600}$ of cell suspension in Z-buffer; $T$: the reaction time in minutes, 15 min; $V$: the volume used in this assay, 1.144 ml. The OD value was further converted to β-gal activity according to a standard curve created in each experiment.

**Fluorescence-based β-galactosidase detection assay.** The levels of active β-Gal in living yeast cells were monitored by SPiDER-βGal[41] (Dojindo Laboratories, Kumamoto, Japan), a fluorogenic β-galactosidase substrate probe with cell permeability. Cells (if needed, they were suspended in 50 mM citrate buffer (pH 4.0) to inactivate the β-Gal, and the buffer was immediately replaced with PBS (pH 7.4)) were suspended in 0.1–1 μM SPiDER-βGal in PBS (pH 7.4) at 30 °C for 30 min, and then washed twice with PBS. Fluorescent intensities inside the cells were acquired using the BZ-X700 system (Keyence Corp., Osaka, Japan).

**Expression and purification of recombinant TaqI.** Recombinant NLS-TaqI was expressed in and purified from *E. coli* Rosetta™(DE3)pLysS (Novogene, Chula Vista, California, USA) transformed with p*TaqI*[16], in which a TaqI-coding DNA fragment amplified from the *T. thermophilus* HB8 (TaKaRa, Shiga, Japan) genome tagged with NLS (Used primers are listed in Supplementary Table 4) is cloned into the pET-15b vector[57] (Novagene). Rosetta cells harboring p*TaqI* were cultured in LB medium at 30 °C, and expression of 6× His-tagged NLS-TaqI was induced by 0.1 mM isopropyl β-D-1-thiogalactopyranoside for 3 h. The *E. coli* cells were then suspended in 50 mM 3-morpholinopropane-1-sulfonic acid (MOPS) (pH 7.4) with 0.3 M NaCl, 5 mM imidazole, and 1× EDTA-free protease inhibitor cocktail (Roche Diagnostics, Mannheim, Germany) and sonicated to prepare crude extracts, which were subsequently incubated at 65 °C for 20 min to denature *E. coli*-derived proteins.

The soluble fraction containing 6× His-tagged NLS-TaqI was obtained from the extracts by TALON Metal Affinity Resin (TaKaRa Bio/Clontech, Mountain View, CA, USA): the resin column loaded with the extracts were washed with the above-mentioned MOPS buffer, and 6× His-tagged NLS-TaqI was eluted by 50 mM MOPS (pH 7.4) with 0.3 M NaCl and 200 mM imidazole. The imidazole-eluted fraction was concentrated, and the buffer was substituted with 10 mM phosphate buffer (pH 7.4) containing 0.3 M NaCl using Amicon Ultra-15 Centrifugal Filter Units 10 kDa cutoff (Merck Millipore Ltd., County Cork, Ireland). The quantity and quality of the purified proteins were verified by Bradford assay and SDS-PAGE (Supplementary Fig. 4a).

In order to verify the nuclease activity of recombinant NLS-TaqI, the obtained TaqI as well as the commercially available restriction enzyme TaqαI (New England Biolabs Inc., Ipswich, MA, USA) was incubated with the pBluescript II SK(+) vector at 37 °C for 5, 20, or 60 min, and DNA digestion was assessed by 1% agarose gel electrophoresis (Supplementary Fig. 4b). Furthermore, the in vitro activity of NLS-TaqI exhibited little differences at 37 and 38 °C: the latter condition was used in the in vivo experiments, TAQing2.0 system, in this study (Supplementary Fig. 4c).

**TaqI activation TAQing2.0.** NLS-TaqI introduced into *C. utilis* strain NBRC0988 cells was activated as follows (TAQing2.0). After protein transfection (see above), the recovery step for Cu cells was adopted by supplying fresh YPD medium and culturing cells at 30 °C for 30 min. The cells were then incubated at 38 °C for 90 min to temporarily activate NLS-TaqI, and cultured on YPD agar plates at 30 °C for a few days. Formed colonies were counted to assess viability after the NLS-TaqI activation, or re-streaked onto YPD agar plates to isolate single colonies for subsequent morphological verification using the BZ-X700 system (Keyence Corp., Osaka, Japan).

**Microscopy and analysis.** All microscopic images were acquired by the BZ-X700 system. The excitation/emission wavelengths and dichroic mirror wavelength of the filter were 525 ± 25 nm/605 ± 70 nm and 565 nm for SPiDER-βGal experiments, and 470 ± 40 nm/525 ± 50 nm and 495 nm for Alexa Fluor 488 goat anti-rabbit IgG and GFP. For each replicate 3–10 fields were acquired. The scale bar in all images is 5 μm. The fluorescence intensity was measured and analyzed using the hybrid cell-counting tool (Keyence).

**Pulsed-field gel electrophoresis (PFGE).** To prepare genomic DNA for PFGE, cells were digested with Zymolyase, embedded in agarose, and further treated with Proteinase K/RNase A. By using the CHEF Mapper XA System (Bio-Rad Laboratories, Inc., Hercules, California, USA), electrophoresis was performed at 14 °C for 48 h with a voltage of 3 V/cm, an included angle of 106°, and a switching time of 500 s. The size marker was purchased from Bio-Rad (Catalog #170-3667).

The DNA was stained with ethidium bromide and scanned using the ImageQuant LAS 4000 scanner (Fujifilm Corp., Tokyo, Japan).

**Genomic DNA isolation.** *C. utilis* genomic DNA was extracted from growing cells by using the genomic DNA buffer set (Qiagen, Hilden, Germany) and Genomic-tip 100/G (Qiagen) according to the manufacturer's instruction, and dissolved in 10 mM TE buffer (pH 8.0).

**Genome resequencing of WT Cu.** Genomic DNA of WT Cu was sequenced using PacBio Sequel (Pacific Biosciences, California, USA) and MiSeq (2 × 300 bp; Illumina, Inc., San Diego, California, USA). Library preparation and genome sequencing was examined by a commercial company (GeneBay, Inc., Kanagawa, Japan). For PacBio sequencing, a 20-kb SMRTbell library was prepared and a total of 3580 Mb was sequenced in 390,195 subreads, achieving a contig N50 of 12 Mb. For MiSeq sequencing, a total of 3,082,222 reads were obtained. The reads were assembled using FALCON and FALCON unzip with the option, overlap_filtering–max_diff 100–max_cov 500–min_cov 5, after filtering the short reads with cutoff options of cutoff = 5000, cutoff_pr=12000. Pilon v1.22[58] and BWA v0.7.12[59] were used to correct errors in FALCON assemblies. The hybrid assembly of PacBio and Illumina sequences was also performed by MaSuRCA v3.2.4[47] with options of GRAPH_KMER_SIZE = 117, cgwErrorRate=0.15, obtaining 30 contigs with a total length of 14,097,946 with an N50 contig size of 1,821,643. The completeness of these assembled contigs was assessed and polished using BUSCO v3.0.2[60] (Supplementary Table 2). The contigs from FALCON and MaSuRCA were assembled manually using genomic similarity search tools, YASS[49] and BLAST + v2.2.29[48]. Telomeric repeat sequences were detected at each end of the contigs to define the assembled contig as a chromosome. Sequencing data were visualized by IGV v2.8.9. The sequence data of the assembled WT Cu reference genome are available at DDBJ Sequence Read Archive (DRA) database under accession numbers DRA012057.

**Mutation and rearrangement detection of TAQed mutants.** Genomic DNA of WT, the two TAQed mutant strains AG4×5 and AG9×5, which were the products of 5 rounds of passive culture of AG4 and AG9, were sequenced using NovaSeq (2 × 150 bp; Illumina).

Library preparation and genome sequencing services were provided by GeneBay. More than $13.8 \times 10^6$ reads were obtained for each strain by NovaSeq. The rearrangement sites were searched according to our previously-described methods[16]. Chromosomal rearrangements and large deletions with chimeric read junctions were detected from the soft-clipping information of BWA-mapping as previously described[16]. These junction sites were confirmed by targeted PCR experiments, where primer sets used are listed (Supplementary Table 4). Small variants, such as SNVs and InDels, were detected by GATK v.4.1.4.1[61] (Supplementary Table 5). Homologous rearrangements, such as gene conversions and break-induced repairs, were detected from the SNV ratio of the mutants to the WT. The sequence data of the WT, AG4x5, and AG9x5 are available at DRA database under accession numbers DRA012846.

**Total RNA isolation.** Cu cells of WT, AG4x5, and AG9x5 were cultivated at 30 °C in YPD. After harvested at a growing phase, total RNA was extracted using acidic phenol pre-heated at 65 °C, and isolated by RNeasy Mini Kit (Qiagen) according to the manufacturer's instruction. Total RNA samples were dissolved in RNase-free distilled water.

**RNA-seq experiments for WT Cu and TAQed mutants.** We used 1 μg of total RNA sample and performed two biological replicates for RNA-seq experiments.

Strand-specific RNA-seq libraries were generated using NEBNext Poly(A) mRNA Magnetic Isolation Module and NEBNext Ultra II Directional RNA Library Prep Kit (New England Biolabs Inc.). Sequencing was performed on an Illumina platform (2 × 150 bp; Illumina) by Macrogen, Inc., and from each sample 5–6 Gb clean data and 31–41 Mb reads were obtained (Supplementary Table 1b). Raw data reads were trimmed using Trimmomatic-0.36 tool[62], and mapped to the *C. utilis* NBRC0988 reference genome (see "Genome resequencing of WT Cu" section) using STAR-2.7.9a tool[63] (mapping rate 93.47%). The featureCounts program (subread v2.0.1)[64] was applied to count mapped reads per gene, and differentially expressed genes (DEGs) were detected by adopting the following conditions; average logCPMs are larger than 0.5 and the absolute values of $Log_2FC$ are larger than 1.3. DEGs were indicated as red dots in MA plots (Supplementary Fig. 5). The raw sequence and gene expression data from WT Cu, AG4x5, and AG9x5 are available at DRA and Genomic Expression Archive (GEA) database under accession number DRA012997 and E-GEAD-459, respectively.

**Statistics and reproducibility.** The statistical analysis was performed by one-tailed Welch's *t*-test from at least three biological replicates. Sample sizes are described in detail in figure legends. The *P* value cutoff was 0.05. *, **, and *** in the figures refer to *P* values <0.05, <0.01, and <0.001, respectively. Reproducibility was confirmed by performing at least two independent experiments. All

microscopic images were acquired 3–10 fields for each replicate, and each quantitative experiment was analyzed using more than triplicate fields.

**Reporting summary**. Further information on research design is available in the Nature Research Reporting Summary linked to this article.

## Data availability

Sequencing data are deposited in the DDBJ/EMBL/GenBank database under accession numbers AP024664-AP024669 and DDBJ Sequence Read Archive database under accession number DRA012057, DRA012846, and DRA012997. Gene expression data of RNA-seq experiments are deposited in DDBJ Genomic Expression Archive (GEA) under accession number E-GEAD-459. Source data for underlying the graphs and plots in the main figures are provided in Supplementary Data. An uncropped agarose gel image of pulse-field gel electrophoresis (Fig. 3e is in Supplementary Fig. 6). All other data are available from the corresponding authors upon reasonable request.

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

## Acknowledgements

English language editing was performed by Japan Translation Center, Ltd (https://www.jtc.co.jp/english/). This work was supported by AMED Grant Number JP20wm0325003 and Japan Science and Technology Agency (JST) CREST JPMJCR18S3 to K.O.

## Author contributions

T.N., T.Y., and K.O. conceived the study; A.H.O. and N.M. designed and performed the genomic analyses. T.N. and T.Yamada determined the conditions for the TAQing2.0 in Sc, T.Y. and M.I. determined those in Cu. T.Y. and T.N. generated the torula yeast mutants and conducted physiological analyses. M.T., Y.Y., and T.Y. performed RNA-seq experiments. All the experiments were performed under the supervision of K.O. K.O., T.Y., A.H.O., and T.Yamada wrote the manuscript, and all authors read and commented on the manuscript.

## Competing interests

The authors declare no competing financial interests but the following competing non-financial interests: This work was partly supported by Mitsubishi Corp. Life Sciences, Ltd. (Japan). T.Y., Y.Y., M.I., and N.M. are employees of Mitsubishi Corp. Life Sciences, Ltd. Patent applications have been filed for the technology described in this publication. T.N., T.Y., M.I., K.O., and N.M. are named as the inventors of these patents. The remaining authors declare no competing interests.
