## [Peer Review File · Communications Biology]

Reviewers' comments:

Reviewer #1 (Remarks to the Author):

In the manuscript by Yasukawa et al. titled "TAQing2.0: Genome reorganization of asexual industrial yeasts by direct protein transfection", the authors upgraded and optimized a method to engineer yeast genome by introducing TaqI restriction enzyme. They performed detailed tests on the conditions to deliver the protein into the yeast. Eventually, they obtained strains which have been modified and produced specific phenotypes. Overall, this is a well-executed study and will be interesting to people in the field of whole genome engineering. I have a few concerns:

- 1) The efficiency of β -Gal seems quite low when comparing the one in Fig1b with or without Trypsin treatment. Are there any improvements using the new conditions such as culture medium, reagents etc?
- 2) Is the deliver method toxic to the yeasts? How many(portions) cells will die after the treatment?
- 3) Are there any gene transcription changed in AG4 and AG9. I will recommed to perform an RNA-seq experiments, which will help to understand the mechanisms of the phenotypes observed

Reviewer #2 (Remarks to the Author):

In this Ms, the authors report a protocol of protein transfection in yeast strains and its use to stimulate genome-wide chromosome rearrangements upon delivery of a foreign TaqI endonuclease.

The significance and the need of this development is well argued in the Introduction, explaining why protein delivery overcomes nucleic acid engineering limitations for biotechnological applications, related to the acceptance of genetically modified organisms, including yeasts. To be exhaustive, the authors should mention the transformation-free yeast-specific process of meiotic reversion that allow to efficiently generate recombination-dependent genome diversification of sterile polyploid yeasts, as recently published in Microbiological Research (Serero et al, 2021).

The first part of the results concerns the development of a yeast protein transfection method, inspired from mammalian cell protocol but optimized for yeast cells to overcome the cell wall barrier. Step by step, convincing experiments initially using quantitative (colorimetric, fluorescence based) reporter assays demonstrate the intracellular localization and activity of the transfected proteins in two yeast strains, *S. cerevisiae* and the evolutionary distant *Candida utilis*, widely used in industrial fields. Parallel analyses of these yeasts report routes of optimization (for example media and growth conditions parameters) suggesting wide application of the proposed and well described methods.

The second part of the results concerns the application of the protein transfection methods to induce genome wide chromosomal rearrangements, using a previously described TaqI endonuclease induction upon nucleic acid transformation and heterologous gene expression. The long and short reads NGS data and efforts to resolve the complex polyploid genome of *C. utilis* clearly demonstrate that TaqI transfection is active and triggers massive chromosomal rearrangements with example of beneficial phenotypic outcomes.

One minor weakness is the PFGE (Fig. 3e). The signal is rather fuzzy and the expected chromosomal bands are not very clear. A better gel, if necessary different gels with various chromosome length resolution, would definitively ascertain the NGS derived conclusion of a 5+1 chromosome genome (add a length scale on the PFGE figure) Perhaps related, to the observation of smeared DNA on this PFGE gel, we would like to know to which extent TaqI is definitively inactive at 30°C and is stable/degraded overtime and cell divisions.

Altogether, the work is well executed and of general interest, certainly of potential application in various yeasts and beyond.

Reviewer #3 (Remarks to the Author):

Dear editor:

I would suggest to publish this article only if major improvements are made. This is an interesting study and the method could benefit the future genome engineering. However, the article is a bit unorganized. At the beginning the authors mentioned about developing system in both *S. cerevisiae* (Sc) and *C. utilis*38 (Cu). Then the rest of the results are mainly focused on Cu. There is no clear reasoning to transit into Cu which needed to be emphasized in the revised manuscript. In addition, the authors have obtained the genome sequence of Cu, is it possible to compare the hypothetic ratio of the cutting efficiency to the experimental data? What can be done to improve the efficiency of the genomic rearrangements in the future?

Major issues

Line 88 The TAQing system would be effective even in many non-conventional yeasts, such as brewer's yeast, sake yeast, and industrial yeast (*Candida utilis*). Do brew's yeast, sake yeast and industrial yeast all refer to *Candida utilis*? Or they refer to other types of yeast respectively?

Line 118 What is Xfect method referring to? What is its relationship to CPP? This should be stated earlier than in line 152.

Line 249-262 There is no reference cited nor any data presented. Thus, it is difficult to understand how and what conclusion the authors tried to support from.

Line 263-279 As it defined in the previous paragraph, the TAQing system is a vector based engineering method. It is very confusing to mentioned "TAQing system by using the CPP method in Cu." which is refer to TAQing2.0 system in the later. Need to rewrite the sentence to make it clearer.

Minor issues

Line 509-512 The authors test the NLS-Taql and TaqI activity at 37 °C in vitro but incubate the cells at 38 °C to activate the NLS-Taql. Why don't you incubate at 37 °C as well?

Line 516-518 The cells ...activate NLS-Taql, and cultured on YPD agar plates for a few days. Did you culture the agar plate at 30 °C? Or at other temperatures?

Line 532was performed at 14°C for 24 hrs with a voltage of 6 V/cm, an included angle of 120 °C

Sincerely,

Huanming Yang

For Reviewer #1

In the manuscript by Yasukawa et al. titled “TAQing2.0: Genome reorganization of asexual industrial yeasts by direct protein transfection”, the authors upgraded and optimized a method to engineer yeast genome by introducing TaqI restriction enzyme. They performed detailed tests on the conditions to deliver the protein into the yeast. Eventually, they obtained strains which have been modified and produced specific phenotypes. Overall, this is a well-executed study and will be interesting to people in the field of whole genome engineering.

We are grateful for your positive comments.

I have a few concerns:

1) The efficiency of β -Gal seems quite low when comparing the one in Fig1b with or without Trypsin treatment. Are there any improvements using the new conditions such as culture medium, reagents etc?

Yes, the efficiency of protein transfection was tuned out to be greatly improved by altering the buffer conditions. After Supplementary Fig. 3(f), we employed the improved buffer conditions: the usage of Tris-HCl (pH 9.0) or MES (pH6.0) instead of PBS for *S. cerevisiae* (Sc) and *C. utilis* (Cu), respectively. We described this point more clearly in the revised manuscript (p6, Line 192-193).

2) Is the deliver method toxic to the yeasts? How many(portions) cells will die after the treatment?

For this question, we obtained new results indicated below (Fig. A).

In summary, we did not detect any toxicity of the treatment of Xfect reagent. We added these data in the text (p5, Line 141-142) and Supplementary Fig. 1.

Fig. A. Viability of Sc and Cu in the presence or absence of the treatment of Xfect reagent

(n=3, n.s., not significant)

3) Are there any gene transcription changed in AG4 and AG9. I will recommend to perform an RNA-seq experiments, which will help to understand the mechanisms of the phenotypes observed

We conducted RNA-seq experiments and compared genome-wide expression between wild type cells and TAQed mutants (AG4 and AG9) with altered flocculation ability (MA-plots in Fig. B). We detected many differentially expressed genes (DEGs). When we focused on genes related with flocculation phenotypes, we observed the remarkable increase in expression of *CuFLO1* and *CuFLO5* genes (Table A and Fig. B), which is consistent with the observed hyper-flocculation phenotypes of AG4 and AG9 strains. We described these points in the text (p9, Line 297-299) and Supplementary Fig. 5 and Supplementary Table 1(a).

Fig. B. MA-plots of genome-wide gene expression data, AG4/WT (left), AG9/WT (right). Red dots refer to genes whose average logCPMs are larger than 0.5 and the absolute values of Log₂FC are larger than 1.3. A green and a blue dot indicates *CuFLO1* and *CuFLO5* gene, respectively.

Table A Expression changes of some flocculation-related genes and housekeeping genes (*ACT1*, *RPL3*)

	Cu ID	Amino acid sequence similarity to Sc counterparts	AG4 / WT		AG9 / WT	
	evm.model.		Log ₂ FC	fold changes	Log ₂ FC	fold changes
FLO1	ChrV.357	little	3.05	8.3	4.07	17
FLO5	ChrV.499	42.9%	1.12	2.2	0.81	1.8
	ChrIII.223	33.2%	4.25	19	3.90	15
FLO8	ChrI.1188	44.0%	-0.97	0.51	-0.57	0.67

FLO9	-	little	-	-	-	-
FLO10	-	little	-	-	-	-
FLO11	ChrV.545	31.0%	0.12	1.1	-0.19	0.88
ACT1	ChrIII.214	95.2%	-0.15	0.90	0.30	1.2
RPL3	ChrII.429	88.9%	-0.92	0.53	-0.98	0.51

For Reviewer #2:

In this Ms, the authors report a protocol of protein transfection in yeast strains and its use to stimulate genome-wide chromosome rearrangements upon delivery of a foreign TaqI endonuclease.

The significance and the need of this development is well argued in the Introduction, explaining why protein delivery overcomes nucleic acid engineering limitations for biotechnological applications, related to the acceptance of genetically modified organisms, including yeasts. To be exhaustive, the authors should mention the transformation-free yeast-specific process of meiotic reversion that allow to efficiently generate recombination-dependent genome diversification of sterile polyploid yeasts, as recently published in Microbiological Research (Serero et al, 2021).

We are sorry for not citing this paper. The meiotic reversion method demonstrated by Sereno et al. indeed an important approach, therefore we cited this paper in the introduction part (p.3, Line 76-78).

*The first part of the results concerns the development of a yeast protein transfection method, inspired from mammalian cell protocol but optimized for yeast cells to overcome the cell wall barrier. Step by step, convincing experiments initially using quantitative (colorimetric, fluorescence based) reporter assays demonstrate the intracellular localization and activity of the transfected proteins in two yeast strains, *S. cerevisiae* and the evolutionary distant *Candida utilis*, widely used in industrial fields. Parallel analyses of these yeasts report routes of optimization (for example media and growth conditions parameters) suggesting wide application of the proposed and well described methods.*

*The second part of the results concerns the application of the protein transfection methods to induce genome wide chromosomal rearrangements, using a previously described TaqI endonuclease induction upon nucleic acid transformation and heterologous gene expression. The long and short reads NGS data and efforts to resolve the complex polyploid genome of *C. utilis* clearly demonstrate that TaqI transfection is active and triggers massive chromosomal rearrangements with example of beneficial phenotypic outcomes.*

One minor weakness is the PFGE (Fig. 3e). The signal is rather fuzzy and the expected chromosomal bands are not very clear. A better gel, if necessary different gels with various chromosome length resolution, would definitively ascertain the NGS derived conclusion of a 5+1 chromosome genome (add a length scale on the PFGE figure) Perhaps related, to the observation of smeared DNA on this PFGE gel, we would like to know to which extent TaqI is definitively inactive at 30°C and is stable/degraded overtime and cell divisions. Altogether, the work is well executed and of general interest, certainly of potential application in various yeasts and beyond.

Thank you for very positive comments. We carefully retried various PFGE conditions and finally obtained better gel images (containing DNA size marker fragments) as shown below (Fig. C). We noticed the longest

chromosome in wild type (chromosome I) exhibited smear bands (indicated with an open bar) possibly due to heterogeneity of rDNA repeat number. Instead, we detected shortened chromosomes in AG4 and AG4x5 indicated with red triangles. The AG9 and AG9x5 mutants have elongated chromosome I. Since we detected duplication at a position proximal to rDNA region in AG9x5 (Fig. 4(a)), it is possible that this duplication may affect the rDNA stability. Based on these findings, we believe that the chromosomal sizes estimated by PFGE are consistent with those deduced by the whole genome sequencing (Supplementary Table 3). We altered the PFGE gel image to these new ones in the revised manuscript (Fig. 3(e)).

Fig. C. Improved PFGE images of AG4, AG4x5, AG9, and AG9x5 strains

We also examined the *in vitro* TaqI activity at 30 °C and 65 °C (see Fig. D) using series of diluted TaqI protein. The results demonstrated that the TaqI activity at 30 °C is largely reduced (1/2,000-1/4,000) as compared to that at 65 °C.

Fig. D. Temperature dependence of TaqI activity at 30 °C and 65 °C

Fig. E. Decay of transfected TaqI protein *in vivo*

The arrow and the asterisk indicate positions of 6xHis-tagged TaqI and nonspecific bands, respectively.

We also investigated the decay process of Xfect-transfected 6xHis-tagged TaqI protein in Cu by immunoblotting with anti-His tag antibody (Fig. E). Cu cells were cultured for indicated hours in YPD medium. We estimated the abundance of TaqI by normalizing the TaqI band intensity with the total band intensity. We estimated more than 50% of TaqI was (61%) degraded within 6 hours.

Reviewer #3 (Remarks to the Author):

Dear editor:

I would suggest to publish this article only if major improvements are made. This is an interesting study and the method could benefit the future genome engineering. However, the article is a bit unorganized. At the

*beginning the authors mentioned about developing system in both *S. cerevisiae* (Sc) and *C. utilis*38 (Cu). Then the rest of the results are mainly focused on Cu. There is no clear reasoning to transit into Cu which needed to be emphasized in the revised manuscript.*

We are sorry for the confusing introduction. In the revised manuscript (p5, Line 127-131), we described the background of this study and the reasons for transition from Sc to Cu more carefully. (We would like to establish a versatile system for multiple yeast species but mainly suitable for Cu. The experiments using Sc is just to initially narrow down the transfection condition applicable for multiple yeast species.)

In addition, the authors have obtained the genome sequence of Cu, is it possible to compare the hypothetical ratio of the cutting efficiency to the experimental data? What can be done to improve the efficiency of the genomic rearrangements in the future?

We estimated the number of TaqI recognition sites in Cu genome is 44,336, which is 1.3 times larger than those in Sc. In addition, the viability of triploid Cu at 38 °C (Fig. 3(c), 8.5%) was much lower than that in haploid Sc cells (40%) at the similar condition (Muramoto et al., *Nat. Commun.* 2018), in which about 200 DSBs occurrence were estimated. Therefore, we speculate that Cu treated with TAQing 2.0 should have more DSBs in the condition used in this experiment. About the improvement of rearrangement efficiency, we can use alternative restriction enzymes such as MseI which shows much stronger activity at 30°C according to our new method of Extended TAQing system (Tanaka et al., *The Plant J.* 2019) .

Major issues

Line 88 The TAQing system would be effective even in many non-conventional yeasts, such as brewer's yeast, sake yeast, and industrial yeast (Candida utilis). Do brew's yeast, sake yeast and industrial yeast all refer to Candida utilis? Or they refer to other types of yeast respectively?

We are now trying to apply TAQing2.0 to Sake yeasts which do not have sporulation ability. We have been obtaining preliminary data, but completion of this project needs substantial time. Alternatively, we indicated phylogenetic trees of those non-conventional yeasts (see Supplementary Fig. 6). Phylogenetic analysis represented sake, brewer's yeasts, and Sc were evolutionally close relationship among them. In this study, TAQing2.0 was effective between Sc and Cu, so we are now speculating it could be also applicable to those yeasts.

Supplementary Figure 6 A phylogenetic tree of Cu-related nonconventional/conventional yeasts.

The phylogenetic analysis was based on comparison with their 18S rRNA gene sequences (about 1,800 bp), which were obtained by this study (*C. utilis* NBRC0988) or from RNAcentral database [1] (<https://rnacentral.org/>). *S. cerevisiae* S288c (URS00005F2C2D_559292), *S. cerevisiae* Kyokai no. 7 (URS00005F2C2D_721032), *S. pastrianus* (URS00008CAE4A_27292), *Zygosaccharomyces rouxii* (URS00003FC14D_4956), *Kluyveromyces lactis* (URS000080E249_28985). Multiple sequence alignments were calculated using DDBJ ClustalW (ver. 2.1, <https://clustalw.ddbj.nig.ac.jp/>). The phylogenetic trees were drawn by using TreeViewX program (ver. 0.5.0). The scale bar represents substitution ratio of base among sequences.

[1] The RNAcentral consortium. RNAcentral: a hub of information for non-coding RNA sequences. *Nucleic Acids Res.* **47**, 221-229 (2019).

Line 118 What is Xfect method referring to? What is its relationship to CPP? This should be stated earlier than in line 152.

Thank you. We revised this part according to your suggestion (p4, Line 95).

Line 249-262 There is no reference cited nor any data presented. Thus, it is difficult to understand how and what conclusion the authors tried to support from.

We are sorry for this. We cited Muramoto et al., *Nat. Commun.* 2018 regarding this part (p8, Line 242 and 245).

Line 263-279 As it defined in the previous paragraph, the TAQing system is a vector based engineering method. It is very confusing to mentioned “TAQing system by using the CPP method in Cu.” which is refer to TAQing2.0 system in the later. Need to rewrite the sentence to make it clearer.

According to your suggestion, we rewrote this part to avoid confusion (p8, Line 254-255) as follow:
 “a new version of TAQing system using the CPP-mediated protein transfection to Cu.”

Minor issues

Line 509-512 The authors test the NLS-Taql and Taql activity at 37 °C in vitro but incubate the cells at 38 °C to activate the NLS-Taql. Why don’t you incubate at 37 °C as well?

We are sorry for this confusing description. We first used 38 °C the *in vivo* experiments, but we unthoughtfully examined the *in vitro* activity at 37 °C. As indicated below (Fig. F), we detected only neglectable differences in the activity of TaqI at 37 °C and 38 °C. We added the results in the revised manuscript (p15, Line 502-504) and Supplementary Figure 4(c).

Fig. F. Comparison of *in vitro* activity of TaqI at 37 °C and 38 °C

Line 516-518 The cells ...activate NLS-Taql, and cultured on YPD agar plates for a few days. Did you culture the agar plate at 30 °C? Or at other temperatures?

We incubated the agar plate at 30 °C. We added this in the revised manuscript (p15, Line 510 and p24, Line 806).

Line 532was performed at 14°C for 24 hrs with a voltage of 6 V/cm, an included angle of 120 °C

Sincerely,

Huanming Yang

Thank you for letting us this typo. We corrected this in the revised manuscript (p16, Line 525-527).

REVIEWERS' COMMENTS:

Reviewer #1 (Remarks to the Author):

I don't have further comments

Reviewer #2 (Remarks to the Author):

The few concerns of this reviewer, in particular the PFGE gel, have been fully taken into account in the revised version.